# A SSM IS POLYMERIZED FROM MULTIVARIATE TIME SERIES

## ABSTRACT

For multivariate time series (MTS) tasks, previous state space models (SSMs) followed the modeling paradigm of Transformer-based methods. However, none of them explicitly model the complex dependencies of MTS: the Channel Dependency variations with Time (CDT). In view of this, we delve into the derivation of SSM, which involves approximating continuously updated functions by orthogonal function basis. We then develop Poly-Mamba, a novel method for MTS forecasting. Its core concept is to expand the original orthogonal function basis space into a multivariate orthogonal function space containing variable mixing terms, and make a projection on this space so as to explicitly describe the CDT by weighted coefficients. In Poly-Mamba, we propose the Multivariate Orthogonal Polynomial Approximation (MOPA) as a simplified implementation of this concept. For the simple linear relationship between channels, we propose Linear Channel Mixing (LCM) and generate CDT patterns adaptively for different channels through a proposed Order Combining method. Experiments on six real-world datasets demonstrate that Poly-Mamba outperforms the SOTA methods, especially when dealing with datasets having a large number of channels and complex correlations. The codes and log files are in the supplementary.

## 1 INTRODUCTION

State space models (SSMs) Gu et al. (2021) Gu & Dao (2023) are subquadratic-time foundational architectures compared with Transformers Vaswani et al. (2017), and shows great performance with approximately linear complexity on long-range dependency tasks. Previous studies Zhang et al. (2023) Wang et al. (2024) Ahamed & Cheng (2024) attempted to employ SSM for Multivariate Time Series Forecasting (MTSF), they all follow the Transformer-based MTSF modeling paradigm: learning dependencies between temporal tokens Zhou et al. (2021) Wu et al. (2021) Zhou et al. (2022) Nie et al. (2022), Channel tokens Liu et al. (2023) and their concatenation Zhang & Yan (2022). However, the special complex dependency pattern of MTS is the Channel Dependency variations with Time (CDT), none of these methods explicitly depict it.

It is inappropriate to directly model the CDT because it not only greatly increases complexity when calculating the dependency between temporal tokens of all channels but is also hard to generalize for the scale of most MTS data. We delved deep into the initial development of SSM Gu et al. (2020): real-time approximation of continuously updating function by orthogonal function basis Voelker et al. (2019), and we found that compared with Transformers, SSM has the potential to efficiently and effectively depict the CDT pattern.

We propose an improved SSM for MTSF tasks named Poly-Mamba. Based on the dynamical equation of its fitting coefficient, we propose three operations, Multivariate Orthogonal Polynomial Approximation (MOPA), Linear Channel Mixing (LCM) and Order Combining, improving the ability of capturing the complex CDT pattern.

The idea of MOPA is to explicitly describe the inter-channel dependency pattern by using the mapping weighted coefficients of the extended multivariate polynomial function basis, which contains variable mixing terms. MOPA is an efficient operation for mapping on the multivariate polynomial basis space after simplification. LCM is a mapping operation used to construct a simple linear correlation between channels. In addition, in order to deal with different MTS scenarios, we propose

Order Combining, which retains the low-order trend information of each channel itself and uses gating Shazeer et al. (2017) to adaptively generate the CDT pattern of each channel.

The architecture of Poly-Mamba for MTSF takes time series patches as tokens and generates corresponding multivariate hidden states. For the multivariate hidden states at each moment, LCM, MOPA, and Order Combining are applied to transform into hidden states with increased inter-channel correlations. Thus, this new SSM process generates a pattern representation of the time-varying dependencies between multi-channel tokens.

Our contributions are as follows:

- We delve into the derivation of SSM, improve it for MTSF, and propose a new model named Poly-Mamba, which explicitly models channel dependency variations with time.

- We develop MOPA, an efficient method based on the idea of extending the orthogonal polynomial function basis space to a multivariate orthogonal polynomial basis space and explicitly characterizing channel dependencies with the mapped weights.

- We propose LCM to handle the simple linear correlation between channels. Additionally, we propose Order Combining, retaining the trend information of each channel itself and adaptively learning different types of CDT patterns.

- We conduct extensive experiments on six real-world datasets. The experimental results show that our Poly-Mamba outperforms the SOTA methods for the MTSF task, proving that the improved SSM can effectively capture the complex dependencies of MTS.

## 2 PRELIMINARY

In Section 2.1, we first briefly review the derivation of SSM dynamics. Subsequently, in Section 2.2, we provide a description of multivariate Legendre polynomials.

### 2.1 SSM

The essence of the state space model is to regard sequence pattern learning as an online function estimation problem. The form basis of its dynamic equation was first proposed in HIPPO Gu et al. (2020). It produces operators projecting arbitrary functions onto the space of orthogonal polynomials with respect to a given measure (Equation 1), The closed-form ODE or linear recurrence (Equation 2) allows fast incremental updating of the optimal polynomial approximation as the input function is revealed through time.

$$\arg \min_{c_1, \cdots, c_N} \int_a^b \left[ u(t_{\leq T}(s)) - \sum_{m=0}^N c_n g_n(t) \right]^2 dt \tag{1}$$

$$\begin{aligned} x'(t) &= Ax(t) + Bu(t) \\ y(t) &= Cx(t) + Du(t) \end{aligned} \tag{2}$$

The target function $u(t)$, a continuously collected signal, is approximated by a linear combination of standard orthogonal function basis $\{g_n(t)\}_{n=0}^N$. For this, HIPPO Gu et al. (2020) uses an online function approximation method, that is, mapping each static interval $s \in [a, b]$ to the entire sequence interval $t \in [0, T]$ through the defined Measure. The analytical solution of the combination coefficient $C_n$ is:

$$c_n^* = \int_a^b u(t_{\leq T}(s)) g_n(t) dt \tag{3}$$

Taking the HIPPO Matrix in S4 Gu et al. (2021) corresponding to HIPPO-LegS as an example, that is, using Legendre polynomials $P_n$ and the corresponding standard orthogonal basis $g_n$, satisfying:

$$\int_{-1}^1 p_m(s) p_n(s) dt = \frac{2}{2n+1} \delta_{m,n}$$

$$g_n(s) = \sqrt{\frac{2n+1}{2}} p_n(s) \tag{4}$$

In LegS, the Measure is defined as $t_{\leq T}(s) = (s+1)T/2$. It uniformly maps the domain $[-1, 1]$ of $P_n$ to $[0, T]$. By taking the derivative of both sides with respect to $T$ and substituting $t_{\leq T}(s)$, $P_n$ and $P_n'$ into Equation 3 (see details in Gu et al. (2020)), the final ODE form is obtained:

$$\frac{d}{dT}c_n(T) = \frac{\sqrt{2(2n+1)}}{T}u(T) - \frac{1}{T}(-nc_n(T) + \sum_{k=0}^{n}\sqrt{(2n+1)(2k+1)}c_k(T)) \quad (5)$$

Putting all the coefficients $c_n(T)$ together and letting $x(t) = (c_0, c_1, \ldots, c_N)$, the dynamic form of SSM is obtained:

$$x'(t) = \frac{A}{t}x(t) + \frac{B}{t}u(t)$$

$$A_{nk} = -\begin{cases} (2n+1)^{1/2}(2k+1)^{1/2} & if \ n > k \\ n+1 & if \ n = k \\ 0 & if \ n < k \end{cases}, B_n = \sqrt{2(2n+1)} \quad (6)$$

S4 Gu et al. (2021) derives the HIPPO Matrix A as a diagonal matrix and gives the corresponding convolution form for efficient training. Mamba Gu & Dao (2023), introduces a selective mechanism into SSM, enabling it to discern the importance of information like the attention mechanism Vaswani et al. (2017).

### 2.2 MULTIVARIABLE LEGENDRE POLYNOMIALS

In the space $L_2(I)$, we can use univariate orthogonal polynomials to construct a set of multivariate orthogonal polynomials Deutscher (2003). Specifically, consider a set of orthogonal Legendre polynomials $\varphi_n(x_c)$ that are exact up to degree $N$ in a single variable $x_c$. Then, the set of multivariate Legendre polynomials $\varphi_{n_1 \cdots n_C}(x) = \varphi_{n_1}(x_1) \cdots \varphi_{n_C}(x_C)$ in $C$ variables constitutes a set of multivariate orthogonal polynomials on $L_2(I)$. The set of multivariate polynomials $\varphi_{n_1 \cdots n_c}(x)$ in $C$ variables up to degree $n_{deg} = \sum_{c=1}^{C} n_c$ has exactly $n_\Phi = \binom{C + n_{deg}}{C}$ elements.

## 3 POLY-MAMBA

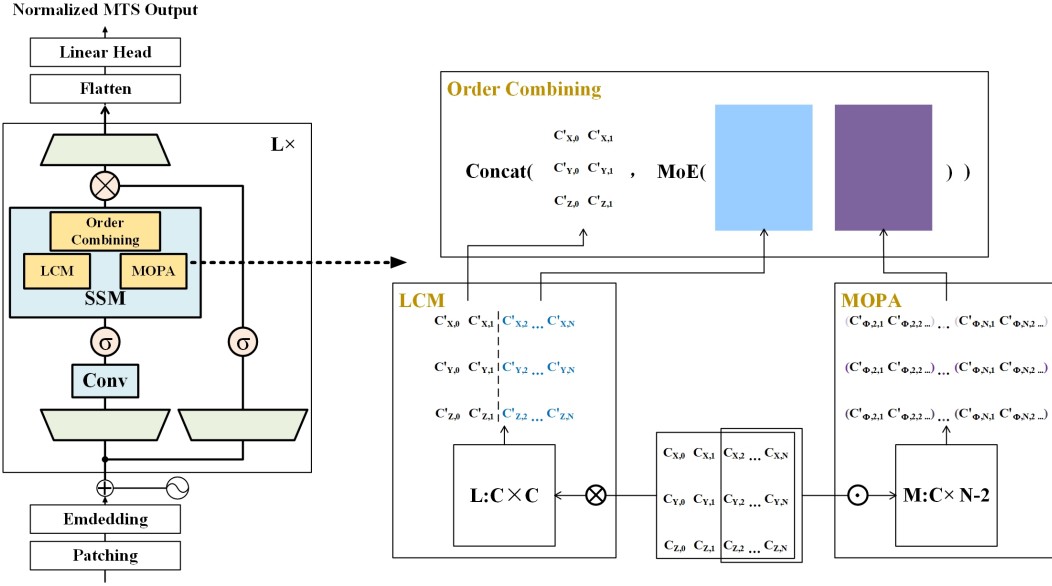

Figure 1: (left) Poly-Mamba architecture. (right) Three operations on coefficients in the improved SSM.

To effectively learn the CDT pattern of MTS, Poly-Mamba models along the temporal direction. We embed sequence segment patches as tokens. This is inspired by the Transformer-based model PatchTST Nie et al. (2022). Treating sequence segments as tokens can be more semantically informative and can effectively reduce complexity, and this is also applicable to the Mamba. As shown in the Figure 1, in addition to the original parameters ($A$, $B$, $C$, $\Delta$) in Mamba Gu & Dao (2023), Poly-Mamba adds operations and corresponding parameters of MOPA, LCM, and Order Combining to the hidden state. The complete process in the form of recurrent is shown in Algorithm 3.3. For the MTSF task, after flattening the encoding result of $L$ layers, a fully connected is used to output the predictions.

## 3.1 MOPA

Our idea is to map the multivariate space containing multivariate mixing terms and describe the complex dependencies between channels with learnable mapping weights. The whole process consists of two steps. The first is to expand the univariate orthogonal polynomial into a multivariate orthogonal polynomial (Equation 7), and obtain the corresponding coefficients. The original basis space with $N + 1$ dimensions is expanded into a basis space with $N_\Phi = \sum_{n_{deg}=0}^{N_{deg}} \binom{C + n_{deg}}{C}$ dimensions.

$$P_{n_1, n_2, ..., n_C}(x_1, x_2, ..., x_C) = P_{n_1}(x_1)P_{n_2}(x_2)...P_{n_c}(x_c)...P_{n_C}(x_C) \tag{7}$$

For example, when the number of channels $C = 2$, a quadratic term $x_1^2$ of the original channel $x_1$ becomes three terms: $x_1^2$, $x_2^2$, and $x_1 x_2$. Here, $x_1 x_2$ represents the mixing term of two variables. The coefficients of different mixing terms represent different degrees of interdependence between channels. The second step is to apply a projection on coefficients of $N_\Phi$ terms:

$$C'_{N_\Phi} = f(Linear(C_{N_\Phi})) = f(W_{N_\Phi}C_{N_\Phi} + B) \tag{8}$$

where $W_{N_\Phi} \in \mathbb{R}^{N_\Phi \times N'_\Phi}$, and an activation function $f$ can be added to make it a nonlinear mapping. Although the weights in the weighted summation of the coefficients of the multivariate orthogonal polynomial function basis can intuitively describe the inter-channel dependence, the two-step calculation plus the linear layer after the dimensional power expansion will greatly increase the complexity.

MOPA makes two simplifications to approximate the above mapping process. First, the dimension $N'_\Phi$ after mapping is set to be equal to $N + 1$, the original dimension of the hidden space. And perform mapping for each order separately, that is, perform weighted summation on the coefficients of all terms of that order. So now the coefficients of each order in the mapped order $N$ are:

$$C'_n = f(\sum_{i=1}^{n_\Phi} w_i C_{n,i} + b_i), n \in \{0, 1, ..., N\} \tag{9}$$

where $n_\Phi = \binom{C + n}{C}$. The first simplification is due to the large amount of repetition and redundancy in previous cross-order linear relationships. Describing the interdependence between channels separately on each order is sufficient. Compared to the complete mapping of the dimension after power expansion, the complexity is significantly reduced.

But the complexity of performing projection on each order of each channel is still too high. MOPA continues to make a simplified approximation. Do not instantiate the weighted summation process and weight $w_i$ for each order. Instead, directly replace the expansion and projection processes with one transformation parameter. Therefore, the coefficient transformation parameter matrix $M$ of size $C \times N$ constructed by MOPA is multiplied by the Hadamard product with the hidden state (coefficient matrix) of size $C \times N$:

$$MOPA(x(t)) = M \cdot x(t) \tag{10}$$

When substituted into Equation 3, the new polynomial Equation 7 and its partial derivative only add the tensor product with polynomials of other variables. Thus, they do not affect the dynamic form of SSM and the parameters in Mamba. As shown in Figure 1 right, in this way, the complex CDT pattern between channels can be concisely and efficiently modeled. Note that the transformation of direct dot product by MOPA is not a completely corresponding method but a simplified variation that can still contain the essence of the original two-step mapping process.

## 3.2 LCM

Different MTS scenarios exhibit varying degrees of CDT patterns. For example, some channels may exhibit a simple linear relationship. Then, for the linear relationship between channels, it is not appropriate for MOPA to capture it with refined high-order channel cross-terms. Therefore, we propose the LCM method. LCM characterizes the linear relationship between channels with a learnable parameter matrix $L$ of size $C \times C$, and multiplies $L$ with the coefficient matrix of size $C \times N$:

$$LCM(x(t)) = L\,x(t) \tag{11}$$

In this way, each channel receives information from other channels and is mapped to the polynomial space linearly related to the channels.

## 3.3 ORDER COMBINING

LCM and MOPA model the simple linear dependency and complex relationships between channels respectively. However, the specific type of channel relationship is unknown in advance. Order Combining adaptively generates the CDT pattern by applying a Gate operation Shazeer et al. (2017) on the outputs of MOPA and LCM:

$$G^{\{2,...,N\}} = softmax(P_L\,LCM^{\{2,...,N\}},\ P_M\,MOPA^{\{2,..,N\}}) \tag{12}$$

where $P_L$ and $P_M$ are the scalar weights. Moreover, in order to maintain the trend (low-order) information of each channel itself, we perform LCM mapping on all terms but only perform MOPA on high-order terms. After performing gate selection on the high-order terms of LCM and MOPA, they are spliced with the low-order terms of LCM:

$$OC = Concat(LCM^{\{0,1\}},\ G^{\{2,...,N\}}) \tag{13}$$

If the overall relationship is linear, then LCM is selected by the gate; if it is a high-order complex relationship, MOPA is selected by the gate, and the coefficients of the low-order terms in front will be adjusted accordingly to adapt to the complex pattern.

---

**Algorithm 1: The procedure of SSM in Poly-Mamba under recurrence form**

| | |
|---|---|
| $[B,C,L,D]$ | **Input**: $Batch(X)$ |
| $[B,C,L,D]$ | **Output**: $Batch(Y)$ |
| | |
| | $Y \leftarrow List[\,]$ |
| | For $t$ in range $(1,\ L)$: |
| $[B,C,D,N]$ | $\quad h_t \leftarrow \overline{A}\,h_{t-1} + \overline{B}\,X_t$ |
| $[B,D,C,N]$ | $\quad h_t \leftarrow Reshape(h_t)$ |
| $[B,D,C,N]$ | $\quad LCM(h_t) \leftarrow L\,h_t$ |
| $[B,D,C,N-2]$ | $\quad MOPA(h_t[2:]) \leftarrow M \cdot h_t$ |
| $[B,D,C,N-2]$ | $\quad MoE \leftarrow softmax(P_L * LCM(h_t)[2:],\ P_M * MOPA(h_t[2:]))$ |
| $[B,D,C,N]$ | $\quad h'_t \leftarrow Concat(LCM(h_t)[0,1],\ MoE)$ |
| $[B,D,C]$ | $\quad y_t \leftarrow C\,h'_t$ |
| $[B,C,D]$ | $\quad y_t \leftarrow Reshape(y_t)$ |
| $[B,C,L,D]$ | $\quad Y \leftarrow Y.append(y_t)$ |
| | End for $t$ |

---

## 4 EXPERIMENTS

We comprehensively evaluate the proposed Poly-Mamba on six real-world datasets, including ETT (consisting of 4 subsets) Zhou et al. (2021), ECL Wu et al. (2021), Exchange Wu et al. (2021), Traffic Wu et al. (2021), Weather Wu et al. (2021) and Solar-Energy Lai et al. (2018). We select seven well-known forecasting models as our baselines, which include: (1) Mamba-based methods: S-Mamba Wang et al. (2024); (2) Transformer-based methods: Crossformer Zhang & Yan (2022), PatchTST Nie et al. (2022), and iTransformer Liu et al. (2023); (3) Linear-based methods: DLinear Zeng et al. (2023), TiDE Das et al. (2023); and (4) CNN-based methods: TimesNet Wu et al. (2022). The experimental setting is identical to that in iTransformer Liu et al. (2023), where the input length $L = 96$ and the output length $T = \{96, 192, 336, 720\}$.

## 4.1 MAIN RESULTS

As is obviously shown in Table 1, by comparing the experimental results with the data visualization of different datasets, we can find that Poly-Mamba has a better prediction performance for scenarios where there is a real-time correlation between channels, such as on Weather. Of course, for scenarios where there is no strong dependence between channels, such as on ETT-m, the prediction effect is on par with that of PatchTST Nie et al. (2022) which uses channel independence.

Secondly, one of the advantages of the state space model is the compression of history. When facing longer prediction interval requirements, Poly-Mamba shows more stable prediction results. Moreover, compared with the S-Mamba Wang et al. (2024) model which is also based on the state space but models along the channel direction, Poly-Mamba shows more excellent prediction stability. For example, by comparing the results of prediction lengths $T = \{336, 720\}$ on the Solar and Weather datasets, we can find that Poly-Mamba's modeling is closer to the complex pattern of them.

Table 1: Main results for the long-term MTSF task. The experimental setting is consistent with iTransformerLiu et al. (2023), where input sequence length $L = 96$ and output sequence length $T = \{96, 192, 336, 720\}$ for all baselines. Avg means the average results from all four prediction lengths. Red value is the best and blue with underline is the second.

| Models | | Poly-Mamba (Ours) | | S-Mamba (2024) | | iTransformer (2024) | | PatchTST (2023) | | Crossformer (2023) | | TiDE (2023) | | TimesNet (2023) | | DLinear (2023) | |
|---|---|---|---|---|---|---|---|---|---|---|---|---|---|---|---|---|---|
| Metric | | MSE | MAE | MSE | MAE | MSE | MAE | MSE | MAE | MSE | MAE | MSE | MAE | MSE | MAE | MSE | MAE |
| ETTm1 | 96 | 0.321 | 0.364 | 0.325 | 0.361 | 0.334 | 0.368 | 0.329 | 0.367 | 0.404 | 0.426 | 0.364 | 0.387 | 0.338 | 0.375 | 0.345 | 0.372 |
| | 192 | 0.365 | 0.387 | 0.368 | 0.385 | 0.377 | 0.391 | 0.367 | 0.385 | 0.450 | 0.451 | 0.398 | 0.404 | 0.374 | 0.387 | 0.380 | 0.389 |
| | 336 | 0.393 | 0.409 | 0.401 | 0.408 | 0.426 | 0.420 | 0.399 | 0.410 | 0.532 | 0.515 | 0.428 | 0.425 | 0.410 | 0.411 | 0.413 | 0.413 |
| | 720 | 0.449 | 0.436 | 0.469 | 0.446 | 0.491 | 0.459 | 0.454 | 0.439 | 0.666 | 0.589 | 0.487 | 0.461 | 0.478 | 0.450 | 0.474 | 0.453 |
| | Avg | 0.382 | 0.399 | 0.391 | 0.400 | 0.407 | 0.410 | 0.387 | 0.400 | 0.513 | 0.496 | 0.419 | 0.419 | 0.400 | 0.406 | 0.403 | 0.407 |
| ETTm2 | 96 | 0.179 | 0.265 | 0.180 | 0.264 | 0.180 | 0.264 | 0.175 | 0.259 | 0.287 | 0.366 | 0.207 | 0.305 | 0.187 | 0.267 | 0.193 | 0.292 |
| | 192 | 0.248 | 0.311 | 0.245 | 0.306 | 0.250 | 0.309 | 0.241 | 0.302 | 0.414 | 0.492 | 0.290 | 0.364 | 0.249 | 0.309 | 0.284 | 0.362 |
| | 336 | 0.318 | 0.356 | 0.309 | 0.347 | 0.311 | 0.348 | 0.305 | 0.343 | 0.597 | 0.542 | 0.377 | 0.422 | 0.321 | 0.351 | 0.369 | 0.427 |
| | 720 | 0.409 | 0.409 | 0.408 | 0.403 | 0.412 | 0.407 | 0.402 | 0.400 | 1.730 | 1.042 | 0.558 | 0.524 | 0.408 | 0.403 | 0.554 | 0.522 |
| | Avg | 0.289 | 0.335 | 0.286 | 0.330 | 0.288 | 0.332 | 0.281 | 0.326 | 0.757 | 0.610 | 0.358 | 0.404 | 0.291 | 0.333 | 0.350 | 0.401 |
| ETTh1 | 96 | 0.384 | 0.404 | 0.382 | 0.401 | 0.386 | 0.405 | 0.414 | 0.419 | 0.423 | 0.448 | 0.479 | 0.464 | 0.384 | 0.402 | 0.386 | 0.400 |
| | 192 | 0.450 | 0.440 | 0.434 | 0.435 | 0.441 | 0.436 | 0.460 | 0.445 | 0.471 | 0.474 | 0.525 | 0.492 | 0.436 | 0.429 | 0.437 | 0.432 |
| | 336 | 0.492 | 0.455 | 0.482 | 0.458 | 0.487 | 0.458 | 0.501 | 0.466 | 0.570 | 0.546 | 0.565 | 0.515 | 0.491 | 0.469 | 0.481 | 0.459 |
| | 720 | 0.484 | 0.475 | 0.481 | 0.480 | 0.503 | 0.491 | 0.500 | 0.488 | 0.653 | 0.621 | 0.594 | 0.558 | 0.521 | 0.500 | 0.519 | 0.516 |
| | Avg | 0.453 | 0.444 | 0.445 | 0.444 | 0.454 | 0.447 | 0.469 | 0.454 | 0.529 | 0.522 | 0.541 | 0.507 | 0.458 | 0.450 | 0.456 | 0.452 |
| ETTh2 | 96 | 0.294 | 0.348 | 0.295 | 0.348 | 0.297 | 0.349 | 0.302 | 0.348 | 0.745 | 0.584 | 0.400 | 0.440 | 0.340 | 0.374 | 0.333 | 0.387 |
| | 192 | 0.373 | 0.392 | 0.376 | 0.397 | 0.380 | 0.400 | 0.388 | 0.400 | 0.877 | 0.656 | 0.528 | 0.509 | 0.402 | 0.414 | 0.477 | 0.476 |
| | 336 | 0.412 | 0.429 | 0.420 | 0.432 | 0.428 | 0.432 | 0.426 | 0.433 | 1.043 | 0.731 | 0.643 | 0.571 | 0.452 | 0.452 | 0.594 | 0.541 |
| | 720 | 0.421 | 0.442 | 0.425 | 0.441 | 0.427 | 0.445 | 0.431 | 0.446 | 1.104 | 0.763 | 0.874 | 0.679 | 0.462 | 0.468 | 0.831 | 0.657 |
| | Avg | 0.375 | 0.403 | 0.387 | 0.405 | 0.383 | 0.407 | 0.387 | 0.407 | 0.942 | 0.684 | 0.611 | 0.550 | 0.414 | 0.427 | 0.559 | 0.515 |
| ECL | 96 | 0.138 | 0.238 | 0.144 | 0.239 | 0.148 | 0.240 | 0.195 | 0.285 | 0.219 | 0.314 | 0.237 | 0.329 | 0.168 | 0.272 | 0.197 | 0.282 |
| | 192 | 0.161 | 0.259 | 0.160 | 0.252 | 0.162 | 0.253 | 0.199 | 0.289 | 0.231 | 0.322 | 0.236 | 0.330 | 0.184 | 0.289 | 0.196 | 0.285 |
| | 336 | 0.172 | 0.272 | 0.176 | 0.271 | 0.178 | 0.269 | 0.215 | 0.305 | 0.246 | 0.337 | 0.249 | 0.344 | 0.198 | 0.300 | 0.209 | 0.301 |
| | 720 | 0.204 | 0.301 | 0.207 | 0.300 | 0.225 | 0.317 | 0.256 | 0.337 | 0.280 | 0.363 | 0.284 | 0.373 | 0.220 | 0.320 | 0.245 | 0.333 |
| | Avg | 0.169 | 0.268 | 0.172 | 0.266 | 0.178 | 0.270 | 0.216 | 0.304 | 0.244 | 0.334 | 0.251 | 0.344 | 0.192 | 0.295 | 0.212 | 0.300 |
| Exchange | 96 | 0.085 | 0.204 | 0.087 | 0.209 | 0.086 | 0.206 | 0.088 | 0.205 | 0.256 | 0.367 | 0.094 | 0.218 | 0.107 | 0.234 | 0.088 | 0.218 |
| | 192 | 0.176 | 0.300 | 0.182 | 0.304 | 0.177 | 0.299 | 0.176 | 0.299 | 0.470 | 0.509 | 0.184 | 0.307 | 0.226 | 0.344 | 0.176 | 0.315 |
| | 336 | 0.332 | 0.418 | 0.336 | 0.421 | 0.331 | 0.417 | 0.301 | 0.397 | 1.268 | 0.883 | 0.349 | 0.431 | 0.367 | 0.448 | 0.313 | 0.427 |
| | 720 | 0.843 | 0.691 | 0.857 | 0.698 | 0.847 | 0.691 | 0.901 | 0.714 | 1.767 | 1.068 | 0.852 | 0.698 | 0.964 | 0.746 | 0.839 | 0.695 |
| | Avg | 0.359 | 0.403 | 0.366 | 0.408 | 0.360 | 0.403 | 0.367 | 0.404 | 0.940 | 0.707 | 0.370 | 0.413 | 0.416 | 0.443 | 0.354 | 0.414 |
| Traffic | 96 | 0.425 | 0.279 | 0.387 | 0.262 | 0.395 | 0.268 | 0.544 | 0.359 | 0.522 | 0.290 | 0.805 | 0.493 | 0.593 | 0.321 | 0.650 | 0.396 |
| | 192 | 0.455 | 0.284 | 0.410 | 0.270 | 0.417 | 0.276 | 0.540 | 0.354 | 0.530 | 0.293 | 0.756 | 0.474 | 0.617 | 0.336 | 0.598 | 0.370 |
| | 336 | 0.467 | 0.292 | 0.425 | 0.277 | 0.433 | 0.283 | 0.551 | 0.358 | 0.558 | 0.305 | 0.762 | 0.477 | 0.629 | 0.336 | 0.605 | 0.373 |
| | 720 | 0.505 | 0.311 | 0.459 | 0.296 | 0.467 | 0.302 | 0.586 | 0.375 | 0.589 | 0.328 | 0.719 | 0.449 | 0.640 | 0.350 | 0.645 | 0.394 |
| | Avg | 0.463 | 0.292 | 0.420 | 0.276 | 0.428 | 0.282 | 0.555 | 0.362 | 0.550 | 0.304 | 0.760 | 0.473 | 0.620 | 0.336 | 0.625 | 0.383 |
| Weather | 96 | 0.155 | 0.204 | 0.172 | 0.214 | 0.174 | 0.214 | 0.177 | 0.218 | 0.158 | 0.230 | 0.202 | 0.261 | 0.172 | 0.220 | 0.196 | 0.255 |
| | 192 | 0.208 | 0.252 | 0.221 | 0.259 | 0.221 | 0.254 | 0.225 | 0.259 | 0.206 | 0.277 | 0.242 | 0.298 | 0.219 | 0.261 | 0.237 | 0.296 |
| | 336 | 0.267 | 0.296 | 0.280 | 0.300 | 0.278 | 0.296 | 0.278 | 0.297 | 0.272 | 0.335 | 0.287 | 0.335 | 0.280 | 0.306 | 0.283 | 0.335 |
| | 720 | 0.344 | 0.347 | 0.358 | 0.349 | 0.358 | 0.349 | 0.354 | 0.348 | 0.398 | 0.418 | 0.351 | 0.386 | 0.365 | 0.359 | 0.345 | 0.381 |
| | Avg | 0.244 | 0.275 | 0.258 | 0.281 | 0.258 | 0.279 | 0.259 | 0.281 | 0.259 | 0.315 | 0.271 | 0.320 | 0.259 | 0.287 | 0.265 | 0.317 |
| Solar-Energy | 96 | 0.200 | 0.231 | 0.208 | 0.241 | 0.203 | 0.237 | 0.234 | 0.286 | 0.310 | 0.331 | 0.312 | 0.399 | 0.250 | 0.292 | 0.290 | 0.378 |
| | 192 | 0.218 | 0.244 | 0.232 | 0.259 | 0.233 | 0.261 | 0.267 | 0.310 | 0.734 | 0.725 | 0.339 | 0.416 | 0.296 | 0.318 | 0.320 | 0.398 |
| | 336 | 0.241 | 0.272 | 0.262 | 0.281 | 0.248 | 0.273 | 0.290 | 0.315 | 0.750 | 0.735 | 0.368 | 0.430 | 0.319 | 0.330 | 0.353 | 0.415 |
| | 720 | 0.245 | 0.279 | 0.258 | 0.284 | 0.249 | 0.275 | 0.289 | 0.317 | 0.769 | 0.765 | 0.370 | 0.425 | 0.338 | 0.337 | 0.356 | 0.413 |
| | Avg | 0.226 | 0.257 | 0.240 | 0.266 | 0.233 | 0.262 | 0.270 | 0.307 | 0.641 | 0.639 | 0.347 | 0.417 | 0.301 | 0.319 | 0.330 | 0.401 |

## 4.2 ABLATIONS

We conducted detailed ablations on each constituent design of Poly-Mamba as is shown in Table 2. The baseline Mamba4TS Hu et al. (2024) for comparison uses the Mamba model with patches as tokens. The results prove that combining LCM and MOPA are important for learning the CDT pattern and neither of them can be omitted. The effectiveness of Order Combining is proved by comparison with Gating only on the outputs of MOPA and LCM (represented as G(LCM,MOPA)) without keeping the low-order trend.

Table 2: Ablations on Poly-Mamba. G(LCM,MOPA) represents Gating only on the outputs of MOPA and LCM, w/o LCM represents only using MOPA and w/o MOPA represents only using LCM.

| Models | | Poly-Mamba | | G(LCM,MOPA) | | w/o LCM | | w/o MOPA | | Mamba4TS | |
|---|---|---|---|---|---|---|---|---|---|---|---|
| Metric | | MSE | MAE | MSE | MAE | MSE | MAE | MSE | MAE | MSE | MAE |
| ETTh2 | 96 | 0.294 | 0.348 | 0.298 | 0.348 | 0.296 | 0.348 | 0.301 | 0.348 | 0.297 | 0.347 |
| | 192 | 0.373 | 0.392 | 0.383 | 0.398 | 0.371 | 0.391 | 0.380 | 0.395 | 0.392 | 0.409 |
| | 336 | 0.412 | 0.429 | 0.419 | 0.429 | 0.418 | 0.429 | 0.422 | 0.431 | 0.424 | 0.436 |
| | 720 | 0.421 | 0.442 | 0.428 | 0.443 | 0.423 | 0.442 | 0.428 | 0.445 | 0.431 | 0.448 |
| | Avg | 0.375 | 0.403 | 0.382 | 0.405 | 0.377 | 0.403 | 0.383 | 0.405 | 0.386 | 0.410 |
| Exchange | 96 | 0.085 | 0.204 | 0.084 | 0.204 | 0.088 | 0.206 | 0.087 | 0.206 | 0.086 | 0.205 |
| | 192 | 0.176 | 0.300 | 0.179 | 0.302 | 0.185 | 0.305 | 0.182 | 0.304 | 0.173 | 0.297 |
| | 336 | 0.332 | 0.418 | 0.331 | 0.418 | 0.334 | 0.419 | 0.348 | 0.428 | 0.340 | 0.423 |
| | 720 | 0.843 | 0.691 | 0.859 | 0.700 | 0.855 | 0.697 | 0.890 | 0.711 | 0.855 | 0.696 |
| | Avg | 0.359 | 0.403 | 0.363 | 0.406 | 0.366 | 0.407 | 0.377 | 0.412 | 0.364 | 0.405 |
| Weather | 96 | 0.155 | 0.204 | 0.163 | 0.210 | 0.164 | 0.211 | 0.164 | 0.210 | 0.175 | 0.215 |
| | 192 | 0.208 | 0.252 | 0.211 | 0.257 | 0.209 | 0.253 | 0.212 | 0.256 | 0.223 | 0.257 |
| | 336 | 0.267 | 0.296 | 0.271 | 0.299 | 0.269 | 0.298 | 0.267 | 0.296 | 0.278 | 0.297 |
| | 720 | 0.344 | 0.347 | 0.350 | 0.354 | 0.347 | 0.351 | 0.350 | 0.350 | 0.355 | 0.349 |
| | Avg | 0.244 | 0.275 | 0.249 | 0.280 | 0.247 | 0.278 | 0.248 | 0.278 | 0.258 | 0.280 |
| ECL | 96 | 0.138 | 0.238 | 0.141 | 0.239 | 0.156 | 0.248 | 0.159 | 0.258 | 0.189 | 0.281 |
| | 192 | 0.161 | 0.259 | 0.163 | 0.259 | 0.168 | 0.264 | 0.177 | 0.275 | 0.194 | 0.286 |
| | 336 | 0.172 | 0.272 | 0.174 | 0.274 | 0.178 | 0.281 | 0.185 | 0.284 | 0.210 | 0.302 |
| | 720 | 0.204 | 0.301 | 0.207 | 0.303 | 0.210 | 0.308 | 0.209 | 0.304 | 0.251 | 0.333 |
| | Avg | 0.169 | 0.268 | 0.171 | 0.269 | 0.178 | 0.275 | 0.183 | 0.280 | 0.211 | 0.301 |

## 4.3 VISUAL ANALYSIS

We carry out visual analysis on MOPA and LCM respectively. We visualize the LCM matrix L, which represents the simple linear dependence among channels. For example, as shown in Figure 2, the linear relationship between channels 'rain' and 'raining' in Weather dataset is more significant than their dependencies on other channels, which is fully in line with the real-world situation.

Although MOPA simplifies the complete polynomial expansion and mapping operations, we can observe the coefficients of different channels before and after the MOPA operation to analyze its effect. As is shown in Figure 3, by comparing with the input curve, it can be found that through the mapping of orthogonal polynomial coefficients by MOPA, the coefficients of relevant variables change corresponding to their roles in the fitting. For example, in ETTh2 dataset, MOPA relatively reduces the absolute values of the coefficients of low order terms and relatively increases the absolute values of the coefficients of high order terms for channels 'MULL' and 'LUFL'. This indicates that MOPA finds out more complex CDT patterns than linear relationships between them, and thus more details, fluctuations or nonlinear relationships in the data are captured. As for the channel 'OT', whose coefficients of all orders were previously in an averaged state, after MOPA, its relationship

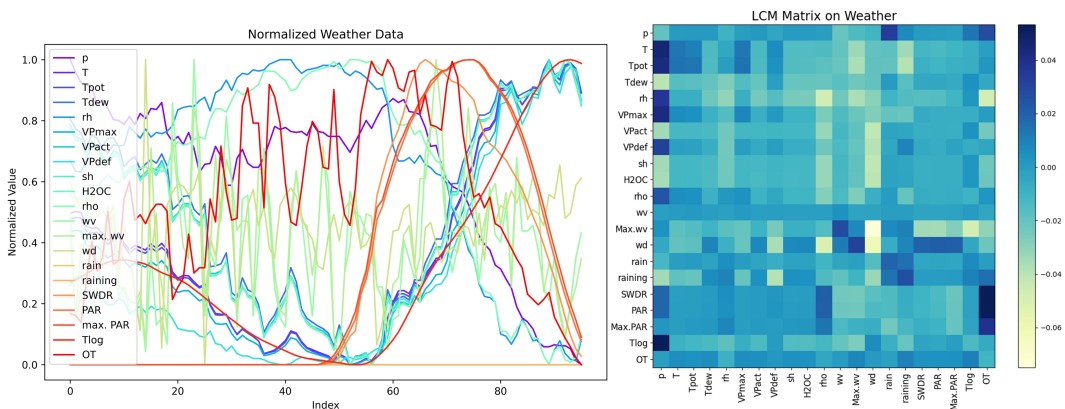

Figure 2: (left) One segment of normalized input of Weather dataset. (right) LCM weight matrix L, which is averaged on all encoder layers.

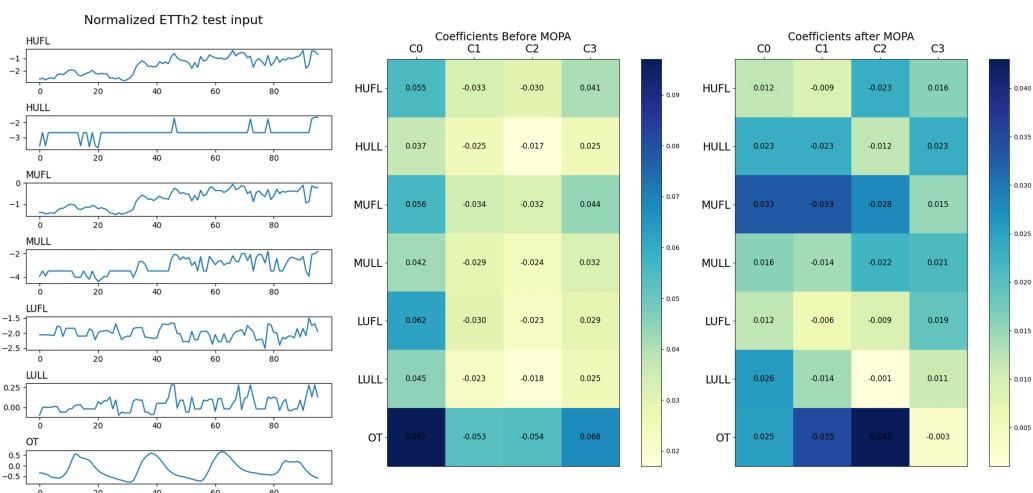

Figure 3: (left) The first normalized test input of ETTh2 dataset. (middle) The corresponding hidden state value, averaged on all encoder layers, before MOPA operation. (right) The corresponding hidden state value after MOPA operation.

with other channels becomes clear and may contain mixed terms, eliminating the weight of its own higher order terms for fitting as before.

## 4.4 EFFICIENCY

Similar to Mamba, we cannot use the convolution form in the early SSM (S4 Gu et al. (2021)) for fast training. The approach in Mamba Gu & Dao (2023), which does not actually materialize the full hidden state $h$, is also applicable to Poly-Mamba. Compared with the efficient implementation in Mamba, Poly-Mamba has only newly added the parameter matrices L, M, and scalar Gating weights P. Together with $(\Delta, A, B, C)$, they are directly transferred from slow HBM to fast SRAM, perform the discretization, additional transformation for CDT learning and recurrence in SRAM, and then write the final outputs back to HBM. The work-efficient parallel scan algorithm of Mamba is also applied for avoiding sequential recurrence. Therefore, as is shown in Table 3, the complexity

Table 3: Model efficiency comparison. All models are under the same setting. The number of Encoder layers is set to 3, batch size is set to 32, embedding dimension is set to 512, and the dimension of the output linear projection is set to 512. All the experiments are implemented in PyTorch Paszke et al. (2019) and conducted on a single NVIDIA TELSA V100 32GB GPU.

| Dataset | Weather (21 Channels) | | | | | | |
|---|---|---|---|---|---|---|---|
| Model | Poly-Mamba | S-Mamba | Mamba4TS | iTransformer | Crossformer | PatchTST | DLinear |
| Memory(GiB) | 0.97 | 0.64 | 0.73 | 0.61 | 1.18 | 1.09 | 0.83 |
| Speed(ms/iter) | 27 | 34 | 23 | 21 | 110 | 31 | 28 |
| MSE | 0.155 | 0.172 | 0.175 | 0.174 | 0.158 | 0.174 | 0.196 |
| MAE | 0.204 | 0.214 | 0.215 | 0.214 | 0.230 | 0.218 | 0.255 |
| Dataset | ECL (321 Channels) | | | | | | |
| Model | Poly-Mamba | S-Mamba | Mamba4TS | iTransformer | Crossformer | PatchTST | DLinear |
| Memory(GiB) | 1.43 | 1.60 | 0.46 | 2.31 | 29.37 | 11.40 | 0.46 |
| Speed(ms/iter) | 677 | 108 | 425 | 108 | 2333 | 507 | 14 |
| MSE | 0.138 | 0.144 | 0.189 | 0.148 | 0.219 | 0.195 | 0.197 |
| MAE | 0.238 | 0.239 | 0.281 | 0.240 | 0.314 | 0.285 | 0.282 |

of Poly-Mamba only slightly increases compared to Mamba. However, for the MTSF task, the prediction performance is significantly improved.

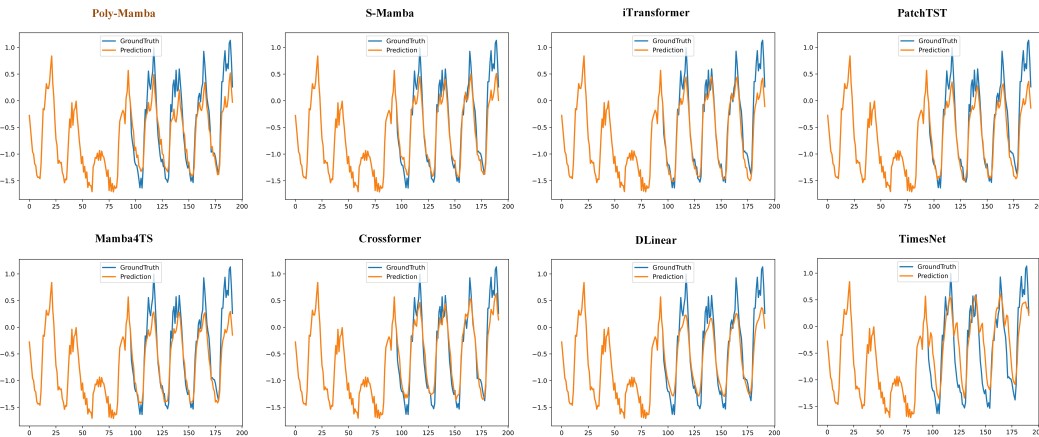

Figure 4: The forecasting of the first test time series from the ECL dataset.

## 5 CONCLUSION

We delved into the derivation of SSM and found that it has the ability to depict the channel dependency variations with time. For Multivariate Time Series (MTS), we transformed the idea of the original SSM's orthogonal function basis for real-time approximation of continuously updated functions into real-time approximation of multivariate functions with continuously updated inter-channel dependencies. Based on this, we proposed the Poly-Mamba model, which contains three modules: MOPA for mapping in multivariate orthogonal polynomial basis space, LCM for simple linear mapping between channels, and Order Combining for retaining trends and adaptive fitting. Experimentally, Poly–Mamba achieves state-of-the-art MTS forecasting performance especially when facing MTS scenarios with a large number of channels and real-time changing inter-correlations. In the future, we will continue to further explore the application of our method in other MTS scenarios and explore large-scale time-series pre-training.

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
