# OpenReview forum: "A SSM is Polymerized from Multivariate Time Series"
_ICLR.cc/2025/Conference — ICLR 2025 Conference Withdrawn Submission_

### Official Review · Reviewer_ASqK · 2024-10-26

**Soundness:** 2
**Presentation:** 2
**Contribution:** 3
**Rating:** 5
**Confidence:** 3

**Summary:**

This paper introduces Poly-Mamba, a novel approach for multivariate time series (MTS) forecasting that addresses limitations in prior state space models (SSMs) by explicitly modeling Channel Dependency variations with Time (CDT). Unlike previous Transformer-based models, Poly-Mamba uses a Multivariate Orthogonal Polynomial Approximation (MOPA) to represent complex dependencies in MTS. It expands the orthogonal function basis to include variable mixing terms, allowing CDT to be described through weighted coefficients. In addition to MOPA which is used to model complex dependencies, it incorporates Linear Channel Mixing (LCM) which models simple relationships. Finally, with Order Combining, the trend (low-order) information is maintained. Experiments across six real-world datasets show that Poly-Mamba surpasses state-of-the-art methods, particularly in handling large, complex MTS datasets.

**Strengths:**

1. This paper presents a well-motivated and effective approach for capturing inter-feature dependencies in state-space models, addressing a gap where many existing models fail to explicitly model these relationships.

**Weaknesses:**

1.	Although the proposed method is grounded in certain theoretical results, I find it challenging for the method to effectively model inter-feature relationships. In MOPA, a weight matrix is element-wise multiplied by coefficients, meaning it cannot explicitly capture inter-feature dependencies. While LCM does capture these relationships, its architectural novelty seems rather limited.
2.	Could you explain how MOPA approximates Equation 8? The formulations appear quite different, so a detailed explanation of the connection would be helpful.
3.	The performance gain in the ablation studies seems somewhat marginal to me.

**Questions:**

See Weaknesses

---

### Official Review · Reviewer_xKbE · 2024-10-30

**Soundness:** 2
**Presentation:** 1
**Contribution:** 1
**Rating:** 3
**Confidence:** 5

**Summary:**

This paper attempts to extend the orthogonal basis projection in the standard SSM to a multivariate orthogonal basis projection and proposes a variant called Poly-mamba, analyzing its practical advantages through various experiments. However, the paper has significant shortcomings in the methodology, evaluation, and writing.

**Strengths:**

The motivation behind the proposed method is meaningful, as the standard SSM is limited to the projection and reconstruction of univariate functions and lacks the ability to model relationships between multivariate time series. Extending the SSM to a multivariate orthogonal polynomial projection provides an **elegant solution**.

**Weaknesses:**

* **W1**: Firstly, the writing of this article needs improvement. The overall impression of this article is rather mediocre, with numerous lines consisting of only one or two words and substantial white spaces. The model diagram and experimental figures are also quite rudimentary. Furthermore, I recommend relocating large tables such as Table 1 & 2 to the appendix, including only summarized results in the main body of the text.

* **W2**: Attention should be paid to certain details. For instance, in line 123, the S4 model utilized a diagonal plus low-rank form of the A matrix, transitioning to using a diagonal A matrix starting from DSS[1] and S4D[2].

* **W3**: This article only conducted standard experiments in the field of time series forecasting, lacking further experimental analysis. For example, (a) performing other time series analysis tasks to explore the potential of Poly-Mamba as a foundational model for time series analysis; (b) conducting experiments with different input lengths to verify that Poly-Mamba still retains its ability to model long-range dependencies. I believe that given the current stage of development in the field of time series analysis, these are essential considerations.

* **W4**: Recent work[3,4,5] has shown that the components of Mamba are detrimental to time series forecasting tasks. The authors should consider improving from the SSM core rather than Mamba.

* **W5**: The performance improvement of Poly-Mamba is marginal, significantly lagging behind existing variants [6,7,8,9] of Mamba for time series forecasting. Despite the paper's claim of modeling inter-variable relationships through multivariate polynomial projection, its effectiveness is even inferior to simply using the SSM core along with an additional channel attention mechanism [10].

* **W6**: The premise of this article is commendable, highlighting the inadequacy of unit polynomial approximations in handling relationships within multivariate time series. However, I believe that the method proposed by the authors does not effectively address this issue: Most modules are designed based on empirical guidance, lacking corresponding theoretical foundations (the theoretical analysis lacks rigorous mathematical proofs and resembles more of a storytelling approach), which is my primary concern regarding this work. For more details, please refer to the **Questions** section.

* **W7**: I think this article should be written following the approach of Hippo, starting from specific orthogonal polynomial families such as Legendre, Chebyshev, Laguerre, etc., to derive the matrix representation of A in the multivariate case, combined with multi-input multi-output SSM systems. Additionally, the article structure could be organized by referencing Koopa[11] and examining the components of time-varying and time-invariant SSMs.

[1] Gupta A, Gu A, Berant J. Diagonal state spaces are as effective as structured state spaces[J]. Advances in Neural Information Processing Systems, 2022, 35: 22982-22994.

[2] Gu A, Goel K, Gupta A, et al. On the parameterization and initialization of diagonal state space models[J]. Advances in Neural Information Processing Systems, 2022, 35: 35971-35983.

[3] Wang S, Li B Z, Khabsa M, et al. Linformer: Self-attention with linear complexity[J]. arXiv preprint arXiv:2006.04768, 2020.

[4] Hou H, Yu F R. Rwkv-ts: Beyond traditional recurrent neural network for time series tasks[J]. arXiv preprint arXiv:2401.09093, 2024.

[5] Sun Y, Dong L, Huang S, et al. Retentive network: A successor to transformer for large language models (2023)[J]. URL http://arxiv. org/abs/2307.08621 v1, 15.

[6] Patro B N, Agneeswaran V S. Simba: Simplified mamba-based architecture for vision and multivariate time series[J]. arXiv preprint arXiv:2403.15360, 2024.

[7] Ahamed M A, Cheng Q. Timemachine: A time series is worth 4 mambas for long-term forecasting[J]. arXiv preprint arXiv:2403.09898, 2024.

[8 ]Zeng C, Liu Z, Zheng G, et al. C-Mamba: Channel Correlation Enhanced State Space Models for Multivariate Time Series Forecasting[J]. arXiv preprint arXiv:2406.05316, 2024.

[9] Behrouz A, Santacatterina M, Zabih R. Chimera: Effectively Modeling Multivariate Time Series with 2-Dimensional State Space Models[J]. arXiv preprint arXiv:2406.04320, 2024.

[10] Hu J, Lan D, Zhou Z, et al. Time-SSM: Simplifying and Unifying State Space Models for Time Series Forecasting[J]. arXiv preprint arXiv:2405.16312, 2024.

[11] Liu Y, Li C, Wang J, et al. Koopa: Learning non-stationary time series dynamics with koopman predictors[J]. Advances in Neural Information Processing Systems, 2024, 36.

**Questions:**

* **Q1**: In the MOPA module, the author simplifies operations using dot product. How can one ensure that the complete multivariate dependency relationships can still be learned?

* **Q2:**: Upon reviewing the code, I discovered that Poly-Mamba does not utilize parallel scanning for accelerated computation but instead computes in a recursive manner. This discovery raises doubts regarding the efficiency analysis presented in Table 3. Why is Poly-Mamba faster than S-Mamba if it does not employ parallel scanning for computation?

* **Q3:**: Poly-mamba still utilizes the standard SSM A matrix, which essentially is a measure kernel function with Laplacian decay. How can one ensure that in this scenario, the dot product operations in MOPA can still model the relationships between multivariate polynomial coefficients effectively?

---

### Official Review · Reviewer_BrqH · 2024-10-31

**Soundness:** 2
**Presentation:** 1
**Contribution:** 2
**Rating:** 5
**Confidence:** 4

**Summary:**

The authors attempt to solve the problem of learning state-space models of multivariate time series data with non-linear interactions amongst the state variables in the observation space. This is done by using orthogonal multivariate Legendre polynomials to capture time-varying correlations amongst the state variables. To mitigate model size increase due to combinatorial explosion, they impose heuristic constraints on the interaction matrix. The model parameters are optimized via a transformer architecture. The authors conclude that their algorithm Poly-Mamba outperforms state of the are methods on, especially when there are correlations amongst the variables.

**Strengths:**

The problem of modeling high-dimensional, multivariate time series data with dynamic correlations is an important, interesting, and long-standing problem with many, many prior works.

**Weaknesses:**

Unfortunately, due to several issues outlined below, this paper is not acceptable for publications in its current form.

Major issues:
	-Most critically, while there may be a preponderance of ‘better performance’ of their models compared to other models, the results are so small as to make the difference practically meaningless. Take, for example, the comparisons made in Table 1 (main results) on the Weather data, which the authors themselves highlight in the results as an example of particular relevance. Here, we indeed see that the author's model performs ‘better’ in the vast majority of cases. However, when one looks at the magnitude of the effect in the third digit, amounting to a 2% -0.2% improvement. I simply do not believe this is a real effect. The insignificance of this result is further highlighted by the plots in Figure 4 (which are never actually discussed), in which many of the model fits to the data are visually indistinguishable from each other, and all seem to make the same quality of errors.
	-I am very concerned about the lack of novelty of the model/method, which seems to be a fairly straightforward improvement of the prior work of Gu, et al., 2020, 2021, 2023. As far as I can tell, it's a very similar model/optimization, now with the utilization of orthogonal Legendre polynomials to represent interactions.
	-Related to my concern about novelty, the author exhibits a general lack of awareness of state-space models in general. For example, in the first line, they cite Gu et al, 2020 for state-space models; however, state-space models have been around for at least 60 years (since before Kalman!).
-Furthermore, to side-step (I don’t think they actually deal with the issue) the curse of dimensionality in terms of high-dimensional interactions amongst variables, they introduce several ad hoc heuristics to simplify the interaction matrix. However, the heuristics they use have little-to-no mathematical support, and it is not discussed what failure modes they may induce. It is not at all clear (at least to me) that the model can support arbitrary interactions of large numbers of variables, thus limiting its utilization. The issue of scaling to large numbers of variables (e.g., 1000s) is never discussed.


Minor issues:
	-the variable “L” is used in many different ways throughout the manuscript, which is confusing.
	-the ablation studies result in similarly small magnitude effects (~2%), making the results unremarkable.
	-writing needs a bunch of work; e.g., capitalization is odd at times, word choice can be confusing, there are omitted words, there are a lot of typos (too many to list here).
	-Several methods that address this problem are not discussed or compared against. For example, the orthogonal stochastic linear mixing model.
	-The most relevant comparison for efficiency seems to be with MambaT5, for which the current algorithm does modestly better but at ~3X memory cost and 50% speed reduction.

**Questions:**

n/a

---

### Official Review · Reviewer_3DXV · 2024-11-03

**Soundness:** 2
**Presentation:** 1
**Contribution:** 2
**Rating:** 3
**Confidence:** 3

**Summary:**

This paper proposes Poly-Mamba, a multivariate time series modeling method that extends the standard SSM to multivariate orthogonal basis projection. The author proposes MOPA, LCM and order combining modules to model the channel correlations. Experiments verify the effectiveness and efficiency of Poly-Mamba in multivariate time series forecasting tasks.

**Strengths:**

The proposed method is meaningful for extending SSM to multivariate time series modeling, especially in the context that most current models including the standard SSM only have univariate modeling capabilities and channel independence performs better.

**Weaknesses:**

- MOPA simplifies the multivariate space mapping process to reduce complexity, but lacks theoretical analysis and approximation errors, which may limit the effectiveness of the proposed method.
- The writing and charts are quite crude and need to be improved.

**Questions:**

- One of the major advantages of SSM is long sequence modeling, but I think the input length of 96 in the experiment may be too short and may not be able to bring out the model's advantage in history compression. How does Poly-Mamba perform when the input sequence length is changed?
- MOPA approximation can reduce complexity, but it may reduce its modeling expression ability. What is its theoretical approximation error in time series modeling?
- Can the algorithmic complexity of Poly-Mamba be given to prove its advantage?

---

### Note · Authors · 2025-01-08

I have read and agree with the venue's withdrawal policy on behalf of myself and my co-authors.